# Predicting Weak-to-Strong Generalization from Latent Representations

## Abstract

AI alignment seeks to align models with human values such as helpfulness and honesty, yet humans may be unable to supervise on tasks exceeding human capabilities. Weak-to-strong generalization (WSG) has been proposed as a proxy for studying this problem, where a weaker model stands in for human supervision and alignment of a stronger model. While prior work provides evidence of WSG success, i.e. the strong model outperforming the weak supervision signal, prior tasks suffer from train-test contamination or rely on oversimplified linear models. We introduce a clean toy-testbed where transformer model pairs are pretrained on different rule variants of Othello and Tic-Tac-Toe, then the stronger model is finetuned on output from the weaker model. It has been hypothesized that WSG works when the strong model learns how to leverage its superior features. While there has been prior theoretical support, we provide empirical evidence for this on transformers. In Othello, the strong student model surpassing the weaker teacher is strongly correlated with having better board representations. Across 111 WSG pairs and 6 game rules, we find a 0.85 Spearman correlation between WSG success and superior board representations in the strong model as measured by linear probes. Our work is a proof-of-concept by analyzing a toy task. By open-sourcing our experiments, we hope to accelerate research on understanding when WSG succeeds.

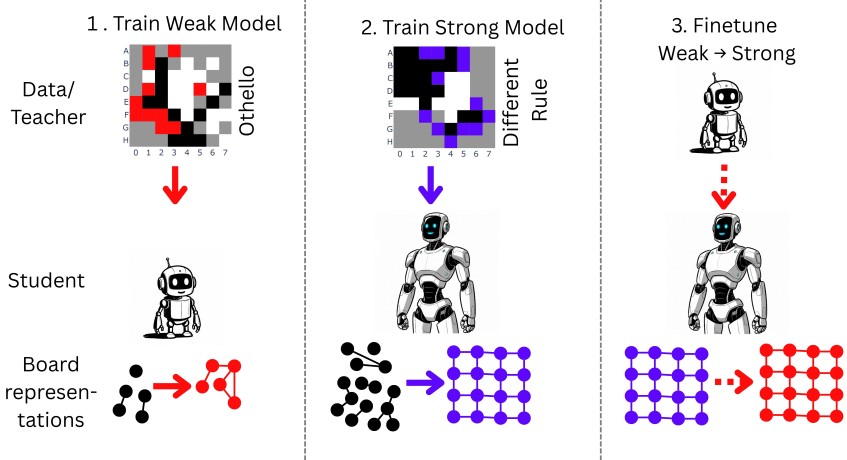

Figure 1: **Overview of our method:** Each column is one step in our pipeline. The top row is the training signal, the middle row the model that gets trained and the bottom row displays how features change during training (structure represents high-quality features, and color the goal). 1. We train a weak transformer on standard Othello, which develops basic board representations (left, red). 2. We train a stronger transformer on modified Othello rules, resulting in superior world modeling capabilities (middle, blue). 3. We fine-tune the strong model using weak supervision, demonstrating that it can exceed the weak teacher's performance when it possesses better initial representations (right). We show that this occurs (almost) if and only if the strong model's board representations are better than those of the weak model. An interpretation is that the strong model learns from the weak model how to utilize its superior features to play standard Othello.

## 1 INTRODUCTION

Current alignment techniques such as RLHF (Christiano et al., 2023) rely on the ability of humans to evaluate AI, i.e. we have a strong teacher (human) that supervises a weaker student (AI). If AI performance surpasses human performance on a task, the direction of supervision stays the same, but the dynamic inverts: A weak teacher (human) has to convey the intended objective to a stronger student (AI). To understand these dynamics, Burns et al. (2023) proposed to study an analogous setup where we train a *weak model* and a *strong model* on two different tasks (called *weak-* and *strong-task*). Then the weak model finetunes the strong model. It was shown on a variety of tasks that **the strong model can surpass the performance of its weaker teacher on the weak task. This phenomenon is called Weak-to-Strong Generalization (WSG)** (Burns et al., 2023; Guo and Yang, 2024; Guo et al., 2024; Somerstep et al., 2024). Burns et al. (2023) proposed the hypothesis that the strong model learns from the weak model how to utilize its superior capabilities (Burns et al., 2023; Shin et al., 2024). If this hypothesis is correct, we should be able to predict when WSG will occur by analyzing how well the strong model's internal representations align with the underlying task structure, compared to its weak supervisor. While there have been theoretical results supporting this hypothesis, no previous evidence on transformers exists. We show that this is true for transformers trained to play variations of the board game Othello. We analyse the toy task of Othello, because it is the first truly pretrain leakage free transformer environment for WSG.

WSG has been studied empirically and theoretically. Burns et al. (2023) using supervision from a human-aligned GPT-2 sized model and achieved performance comparable to GPT-3.5 on NLP classification tasks. One limitation of this class of experiments is that both the task-evaluation and pretraining of the strong model were conducted on human data while future AI won't have examples of superhuman aligned behavior that it can elicit. Theoretical analysis of WSG in simplified settings (e.g., linear models over Gaussian features) has shown that WSG succeeds when the strong model possesses features useful for the weak task (Wu and Sahai, 2024; Shin et al., 2024), that do not support replicating the weak model's errors (Xue et al., 2025; Lang et al., 2024). There has been no prior analysis of what these features actually are. Prior mechanistic interpretability work has analyzed the internal features of models (Elhage et al., 2021), but not around WSG.

We make two main contributions. First, we demonstrate WSG in clean game environments (Othello and Tic-Tac-Toe) by training models on rule variations. Secondly, we investigate WSG with mechanistic interpretability tools in Othello. Specifically, we train a weak transformer to play legal moves in Othello and a strong transformer to play under a modified rule set. After finetuning the strong model using the output of the weak model, the strong model plays legal moves more reliably than the weak model. A similar idea in Tic-Tac-Toe results in the smallest transformer setup with shown WSG. These environments have no pre-train leakage, and we can build on (Nanda et al., 2023; Li et al., 2024a) which showed that in Othello transformers represent the board state as linear directions in the residual stream. We take the accuracy of linear probes on the weak and strong models as a proxy for the strength of board representations. Empirically, the weak-supervision succeeds (almost) if and only if the strong model has better board representations than the weak model.

For 6 different strong rules, we finetune 111 pairs of weak and strong models. We then show that the prediction rule *"WSG occurs if and only if the strong model has better board representations than the weak model."* has a 89% accuracy. We initially established this result using two rules and subsequently confirmed it with an additional four. For each pair we compute the Performance-Gap-Recovered (PGR) (Burns et al., 2023) which is a score for how well the strong model generalized. We show that linear probes trained to predict whether a square is empty, has a stone of the currently moving player or one of the opponent are the most correlated with WSG, as measured by accuracy (88.7%), Spearman rank-correlation (0.850), and $R^2$ (0.298). As an ablation we compare the predictiveness with the difference in model size, loss before finetuning and three different board feature bases (empty/filled), (empty/black/white) and a non-linear transform of the board. Note that (Nanda et al., 2023; Li et al., 2024a) showed that transformers use the empty/ours/opponents stone bases.

As this is the first investigation of WSG using tools from mechanistic interpretability, our work serves as a proof of concept. Othello and Tic-Tac-Toe are new toy-testbeds for this, because they have no pretrain leakage and their rules can be easily modified to test the conditions under which WSG is effective. Our results strengthen the hypothesis, supported by prior theoretical work, that the superior features of the strong model enable WSG.

## 2 BACKGROUND

### 2.1 WEAK-TO-STRONG GENERALIZATION

**Set-up.** Weak-to-Strong Generalization refers both to the phenomenon of weak supervision working, and a setup to experimentally investigate it. First proposed in Burns et al. (2023), a weak model $M_w$ gets trained on a weak task $D_w$ and a strong model $M_s$ on a strong task $D_s$. In the analogy, $M_w$ stands for the human and $M_s$ for the superhuman AI with a different objective. To test if humans can align superhuman AI, we finetune $M_s$ on the output of $M_w$ to get $M_{s \mapsto w}$. A large enough model $M_s$ converges towards $M_w$, but it was observed that early in the finetuning the strong model $M_{s \mapsto w}$ can surpass its weak teacher $M_w$ (Xu et al., 2025; Burns et al., 2023). Experiments usually early-stop the finetuning on the ground truth labels $D_w$, although this is unrealistic since we won't have superhuman ground truth labels.

**Performance-Gap-Recovered (PGR).** In our experiments, we measure performance through the Cross-Entropy loss $CE(M, D)$ for a model $M$ and task $D$. We say WSG occurs when the strong model surpasses its weak teacher, i.e. $CE(M_{s \mapsto w}, D_w) < CE(M_w, D_w)$. We want to know how close the weakly finetuned model $M_{s \mapsto w}$ comes to a strong baseline $M_{sb}$, which is the strong model directly pretrained on the weak task $D_w$. Burns et al. (2023) proposed the Performance-Gap-Recovered (PGR) metric that we adapt for CE-loss. It is 0 if $M_{s \mapsto w}$ matches $M_w$, positive if it surpasses $M_w$ and 1 if it matches $M_{sb}$:

$$PGR(M_w, M_{s \mapsto w}, M_{sb}) = \frac{CE(M_w, D_w) - CE(M_{s \mapsto w}, D_w)}{CE(M_w, D_w) - CE(M_{sb}, D_w)}.$$

### 2.2 MECHANISTIC INTERPRETABILITY OF OTHELLO.

**Linear probe.** The linear representation hypothesis states that models encode many concepts as linear directions in their activation spaces (Park et al., 2024; Elhage et al., 2022; Li et al., 2021; Gurnee et al., 2023; Geva et al., 2021). A linear probe is a linear classifier trained to predict specific properties from a model's intermediate activations. High probe accuracy indicates that the target information is linearly accessible in those representations (Zou et al., 2025; Li et al., 2024b).

**Othello.** We want to analyze whether a superior world understanding of the strong model helps to learn from a weak supervisor. We use the board game Othello, because it was previously shown that a transformer trained autoregressively on Othello moves internally represents the board state and uses it for its output (Nanda et al., 2023; Li et al., 2024a). This is an example of a world model, where to solve a task a model has to gain understanding of the task and not just superficial correlations (Gurnee and Tegmark, 2024; Lovering et al., 2022). Othello is a 2-player game that is played on a $8 \times 8$ chessboard. Players take turns placing a stone on an empty field that flanks opponent pieces in a straight line (horizontally, vertically, or diagonally) between the new stone and an existing friendly stone. After placing, the color of all flanked stones is flipped. Making a legal move therefore depends on the stones in other areas of the board. First, (Li et al., 2024a) showed that there are no linear features that represent whether a field on a board is empty/white/black. Afterwards, (Nanda et al., 2023) showed that the board state is represented as linear directions empty/mine/yours for each of the 64 fields. Using these directions to modify the activations to represent a different board state changed the predicted legal moves accordingly (Nanda et al., 2023; Belinkov, 2021).

## 3 RELATED WORK

**Tasks with shown WSG.** The most used task is NLP classification, either binary (Burns et al., 2023; Ye et al., 2024; Yao et al., 2025; Yang et al., 2024a; Lang et al., 2025) or multiclass (Guo and Yang, 2024). Because these tasks are more similar to knowledge elicitation and use complex LLMs, evaluating the role that world models play in WSG is difficult. Generative tasks have been used for tasks such as answering math questions (Guo and Yang, 2024; Yang et al., 2024b), answering in a specific style (Somerstep et al., 2024) and solving chess puzzles (Burns et al., 2023). These setups have pre-training leakage (Burns et al., 2023), since the strong models dataset $D_s$ contains examples of the weak task $D_w$. But superhuman AI won't have examples of superhuman aligned behaviour

that it can elicit. For example, Burns et al. (2023) used as a weak task $D_w$ chess puzzles where a model learns to predict the best move in a chess puzzle. But the pretraining dataset $D_s$ of the strong model GPT4 contained not only text data but also chess games of players ranked above 1800 Elo. Guo et al. (2024) use a task without pretrain leakage in which they finetune a vision autoencoder for image classification. But the used models are not transformers and it is not a generative task. Our Othello and Tic-Tac-Toe environments are the first generative tasks, where the strong data $D_s$ does not contain examples of the weak task $D_w$. Theoretical analysis of WSG has been mostly focused on linear models over fixed features. These findings often have been empirically validated on Gaussian distributions (Wu and Sahai, 2024; Ildiz et al., 2025; Shin et al., 2024; Charikar et al., 2025) which do not exhibit characteristics such as superposition (Elhage et al., 2022). Xue et al. (2025) empirically tests the role of the strong models features by finetuning a full transformer, while other work has trained only a linear model as the last layer.

**Games in mech-int.**   Board games have been successfully used in mechanistic interpretability before. For chess (Toshniwal et al., 2022), Othello (Nanda et al., 2023; Li et al., 2024a) and Tic-Tac-Toe (Ayyub, 2025) parts of the algorithm were reconstructed. Chess has also been used as a benchmark for interpretability techniques (Toshniwal et al., 2022). We continue this line of work by showing that WSG occurs and can be investigated in Othello and Tic-Tac-Toe.

**Theoretical analysis of WSG.**   Prior theoretical results for simplified scenarios have shown that the features of the weak and strong model influence whether WSG is possible. The strong model ideally has internal representations that are useful for the weak task but do not help replicate the weak model's systematic errors (Xue et al., 2025; Lang et al., 2024; Wu and Sahai, 2024; Mulgund and Pabbaraju, 2025). Data points are needed on which the strong model can learn from the weak model how to utilize its superior features (Shin et al., 2024). Theoretical analysis shows that while extensive fine-tuning eventually causes strong models to converge to weak teacher behavior, models with sufficiently powerful representations can surpass their teachers during intermediate training phases Dong et al. (2025); Xu et al. (2025) Our mechanistic interpretability approach provides the first direct empirical validation of these theoretical predictions using realistic transformer architectures.

# 4    GAME ENVIRONMENT TO STUDY WSG

## 4.1   METHOD

**Othello Data.**   We base our Othello environment on the code of (Li et al., 2024a) and expand it by adding new game rules and the WSG-pipeline. We represent games as token sequences where each of the 60 playable squares corresponds to a token. Games are sampled using 7 different rule variants 1 with moves chosen uniformly at random from all legal moves under that rule. We remove the $\approx 1\%$ of games that end early because no player can move. We split the data into four sets. To prevent data leakage from a model memorizing sequences, we split over the 12 possible combinations of first two moves in Othello and start training on the third move. This split is the same for all rules we train on: for non-Othello rules we sample the first 2 moves from it. We use 26M games for pretraining the weak and strong model and for linear probes (`train`), 13M for the weak model's finetuning phase (`finetune`), 4M for early-stopping (`val`) and 9M for evaluation (`test`).

**Othello Training.**   We then train transformers autoregressively using cross-entropy loss. For example, if we train on the rule `standard` (regular Othello 1), our model predicts the third move F6 of the sequence [F4, F5, F6, G4, ...] given the preceding moves [F4, F5]. Since we sampled the data uniformly over all legal moves, a perfect model should assign for the 4 legal moves {F6, D2, C3, E5} a uniform probability of $1/4$ each. We do not explicitly teach the model any rules of Othello, instead it learns purely from next-token prediction on randomly sampled games. We adopted hyperparameters A.2 from (Li et al., 2024a) where possible. We pretrain with this procedure GPT-2 style transformers of different sizes on the rules `standard`, `bias_clock`, `next_to_opponent` and `no_flipping` as defined in 1. We create two additional models that stay untrained: one with random parameters `untrained` and one with constant parameters `constant_parameters`. Lastly, we take the Chess-transformer from Toshniwal et al. (2022) and train new embedding and unembedding matrices for Othello by keeping all other parameters fixed and training on the `bias_clock` data to obtain `chess`. This idea was previously applied to vision models in LLaVA

Liu et al. (2023). We use 7 different sizes `nano`, `micro`, `mini`, `small`, `medium`, `large`, `huge` A.1 where `huge` is the same architecture as the Othello (Li et al., 2024a) and Chess (Toshniwal et al., 2022) transformers. As a result, we have for all 7 rules each 7 transformers pretrained (only one for `chess`). We define the *weak task* as `standard`, which stands in the WSG analogy for aligned behaviour. Models trained under one of the other 6 rules (*strong task*) represent misaligned AI. We create pairs of *weak models* $M_w$ trained on `standard` and *strong models* $M_s$ trained on a different rule. Then we finetune through CE-loss the strong model $M_s$ on the soft labels of $M_w$, which stands for humans supervising the misaligned AI. This results in $5 \cdot (1 + 2 + 3 + 4 + 5 + 6) + 1 \cdot (6) = 111$ datapoints. Finetuning is early-stopped based on ground-truth labels, which is the standard way in the literature. However, this is a missing piece in the real-world alignment analogy since we might not be able to evaluate superhuman models.

**Tic-Tac-Toe.** Our Tic-Tac-Toe environment and training work similar to Othello. It builds on the Tic-Tac-Toe implementation of Ayyub (2025). The transformer input are game sequences sampled uniformly over all legal moves. Both rules 1 `standard` (the weak task) and `no_diagonals` (the strong task) have different target soft probabilities. Instead of predicting the next token, the model is now required to learn uniform probabilities over the optimal moves under that rule (which we computed using a min-max algorithm). While `standard` is regular Tic-Tac-Toe, in `no_diagonals` a player that completes a diagonal automatically loses 1. The train-test split is again over the first two moves. Since Tic-Tac-Toe is small, it happens that two games from train and test that started differently end up at the same state. To minimize this, we modify the min-max reward to $-1$ for a loss, 0 for a draw, 1 for a win and 2 if one of the winning conditions includes a player's first stone (we splitted over these.). Still, $10\%$ of the training board states are also part of the test set. The reason is that board states with enclosed first placed stones can also occur in the same way in the test set. We run a sweep of $n = 10$ by independently generating the data, splits and model trainings.

Table 1: **Rule Definitions for Othello and Tic-Tac-Toe**

| Rule | Definition |
|---|---|
| *Othello* | |
| `standard` (weak rule) | Only legal Othello moves. Uniform probability. |
| `bias_clock` | Only legal Othello moves. 80% chance of field closest to corner `move-index%4`. 20% uniform $\implies$ strong bias. |
| `next_to_opponent` | Uniform over fields next/diagonal to an opponent piece. Only flips neighbors $\implies$ no long-range dependencies. |
| `no_flipping` | 70% chance of uniform over fields next/diagonal to an opponent piece. 30% chance random field. No stones get flipped. |
| `chess` | Chess model from Toshniwal et al. (2022) adapted to Othello vocabulary with LLaVA (Liu et al., 2023). |
| `untrained` | Strong model is randomly initialized. |
| `constant_parameters` | Strong model starts with all weights and biases set to their mean (except embeddings and first attention module). |
| *Tic-Tac-Toe* | |
| `standard` (weak rule) | Uniform over min-max optimal moves in Tic-Tac-Toe. |
| `no_diagonals` | Uniform over min-max optimal moves for Tic-Tac-Toe if completing a diagonal instantly looses. |

## 4.2 RESULTS

**Othello.** In 2 we can see that for rules more similar to standard Othello, the PGR is positive, i.e. the strong student surpassed its weak teacher. An intuitive ordering of how similar the rules are to Othello also matches roughly how well the generalization happens. Chess did not work, however this might be because the model from Toshniwal et al. (2022) was trained on portable game notation, where a single token does not correspond to a single move.

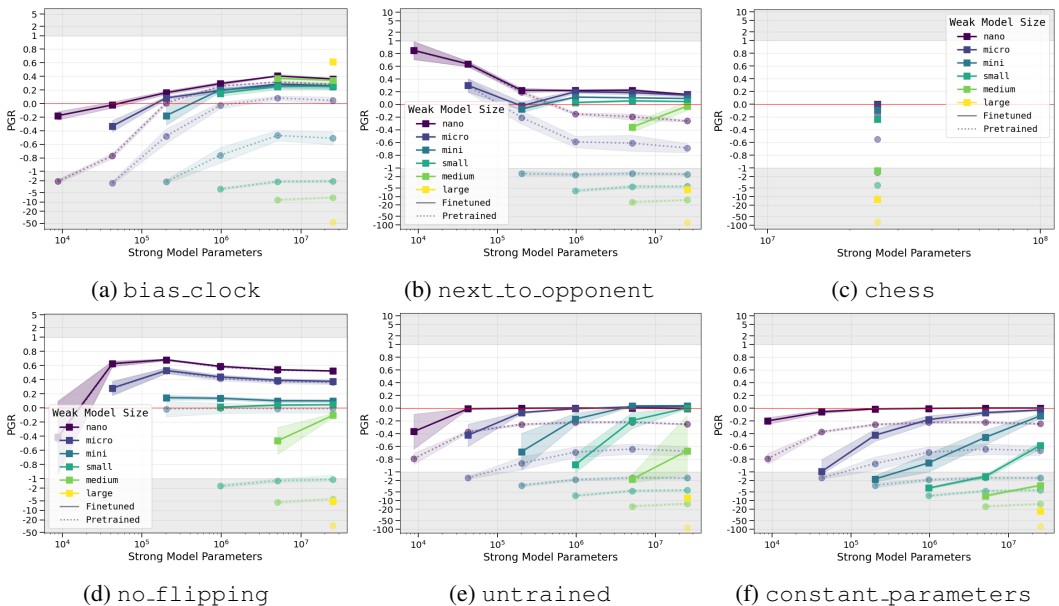

(a) bias_clock  (b) next_to_opponent  (c) chess

(d) no_flipping  (e) untrained  (f) constant_parameters

Figure 2: **Othello: Performance-Gap-Recovered for different model sizes and strong rules**. Each subplot shows the WSG result for one rule variant 1. E.g. in 2a a biased Othello model has to learn to play unbiased. Each square represents a $(M_w, M_s)$ pair where the weak model $M_w$ was trained on standard Othello and the strong model $M_s$ on the modified rule indicated in the subplot title. The x-axis is the strong model size n_params$(M_s)$, and the color is the weak model size (as defined in 3). Finetuning $M_s$ through $M_w$ results in $M_{s \mapsto w}$. The y-axis shows Performance-Gap-Recovered (PGR) 2.1 which is a score that measures the success of WSG. 0 indicates the fine-tuned strong model $M_{s \mapsto w}$ matches the weak teacher's $M_w$ performance, positive values indicate successful WSG where $M_{s \mapsto w}$ surpassed $M_w$. The circular dots are the same metric but computed before finetuning as an ablation to check if the strong model was already performing well on standard Othello before finetuning. WSG success correlates with rule similarity to standard Othello. Rules with minor modifications (bias_clock 2a, next_to_opponent 2b, no_flipping 2d) frequently achieve positive PGR, while more unrelated strong models (chess 2c, untrained 2e, constant_parameters 2f) rarely exceed weak teacher performance. For a sweep of $n = 3$ the mean and its standard errors, i.e. $\mu \pm \sigma/\sqrt{n}$ get displayed.

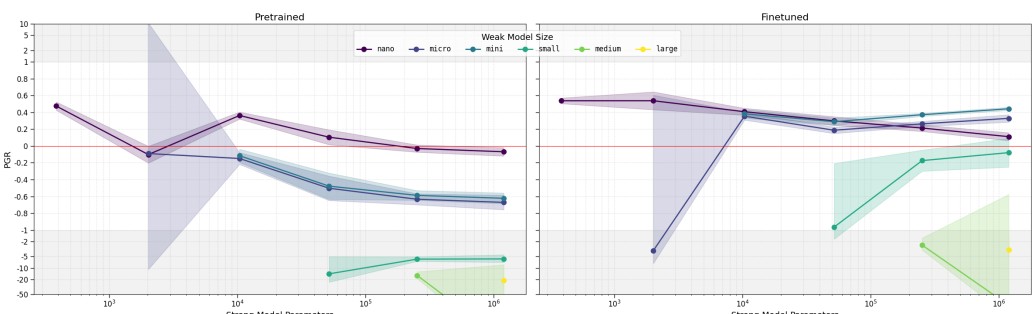

Figure 3: **Tic-Tac-Toe: Performance-Gap-Recovered for different model sizes**. The plot is the same as 2. We plot for a sweep of $n = 10$ the mean and its standard errors, i.e. $\mu \pm \sigma/\sqrt{n}$. For small weak models PGR is positive, i.e. they are surpassed by their stronger students. For large weak models (medium, huge), PGR is negative, because they are already playing close to perfect.

**Tic-Tac-Toe.** In figure 3 WSG success in Tic-Tac-Toe depends on weak model capability. Smaller weak models (nano, small, mini) are consistently surpassed by their strong students, while larger weak models perform too well to leave room for improvement. Except for some outliers, the PGR score is stable across the sweeps.

## 5 PREDICTING WSG THROUGH MECHANISTIC INTERPRETABILITY

### 5.1 METHOD

**Training Linear Probe.** Nanda et al. (2023); Li et al. (2024a) showed a transformer trained on Othello (linearly) represents the board state and uses it. We use linear probe accuracy as a proxy for how well board state information is encoded in the model's representations. Our goal is to test the hypothesis that a difference in the quality of board features between the strong and weak model is related to the strength of WSG.

We use a similar setup to Nanda et al. (2023) and predict for games of the rule `standard` with a linear probe the board state in the basis empty/mine/yours. Concretely: For an input subsequence $x_1, \ldots, x_k$ we take the latent activations at the $k$-th token of layer $l = \text{round\_down}(4/3 \cdot \text{n\_layer})$. This is a vector $a_l \in \mathbb{R}^{\text{d\_model}}$, where d\_model is the dimension of the residual vector. For the board state we define for each of the 64 board squares one target $y_1, \ldots, y_i, \ldots y_{64} \in \{(1,0,0), (0,1,0), (0,0,1)\}$ where $y_i$ is $(1,0,0)$ if the $i$-th field is empty, $(0,1,0)$ for having a stone of "my" color, i.e. of the currently moving player and $(0,0,1)$ if it is the color of the opponent ("yours"). Then, we train 64 linear models of the form $\text{softmax}(W_i \cdot a_l) \approx y_i$ where $W_i \in \mathbb{R}^{3 \times \text{d\_model}}$ using the hyperparameters A.2. The linear probe prediction is the argmax of the softmax probabilities. The reported accuracy score is the mean over all 64 fields $i$ and all subgames of 1k games. An untrained model has 33% accuracy and since fields are more often empty, each trained probe should get $> 46\%$. We obtain for every pretrained and finetuned model $M$ a score LP-acc$(M)$.

**Measure of success.** We want to test the hypothesis that strong models with better board representations than their weak teachers are more likely to exceed the teacher's performance through finetuning. We define $X_i = \text{LP-acc}(M_s) - \text{LP-acc}(M_w)$, where $(M_w, M_s)$ is the $i$-th (weak, strong) model pair. Note that the weak model is smaller than the stronger one. We want to measure the relation between $Y_i = PGR(M_w, M_{s \mapsto w}, M_{sb})$ and $X_i$. Across all $i$, we evaluate a) the sign accuracy $\mathbb{E}[\mathbf{1}_{\text{sign}(X_i)=\text{sign}(Y_i)}]$ where $\mathbf{1}$ denotes the indicator function, b) Spearman's rank correlation coefficient $\rho_s(X, Y) = \rho(\text{Rank}(X), \text{Rank}(Y))$ and c) the coefficient of determination $R^2(X, Y)$. Note that since PGR is a non-linear transformation of model performances, Spearman's rank correlation is more informative than Pearson's, as the former captures non-linear monotonic relationships between $X$ and $Y$ (de Winter et al., 2016). We are using the sign accuracy of $X$ and $Y$ as the simplest possible classification rule that tests the hypothesis that better representations allow the strong student to surpass its weak teacher.

**Ablation.** As an ablation, we repeat the analysis with different definitions of $X_i$ to compare how well WSG success gets predicted by other attributes. First, we test three alternative board state representations. Prior work showed that transformers naturally use an empty/mine/yours basis rather than empty/white/black Nanda et al. (2023); Li et al. (2024a). We also test an empty/filled basis, which captures less strategic information since it only tracks square occupancy. As a negative control, we create a deliberately uninformative feature by applying a random linear transformation $\{-1, 0, 1\}^{64 \times 64}$ followed by a modulo 3 operation to the empty/mine/yours labels in $\{0, 1, 2\}^{64}$, expecting this to show no correlation with WSG success. Finally, we test two potential confounds: whether model size differences $X_i = \log(\text{n\_params}(M_s)) - \log(\text{n\_params}(M_w))$ alone predict WSG success, and whether strong models that already outperform weak models on the target task (before fine-tuning) simply achieve higher PGR scores by default: $X_i = \text{CE}(M_w, D_w)) - \text{CE}(M_s, D_w))$.

### 5.2 RESULTS

**Relationship between Features and WSG.** Table 2 shows that the difference in strength of linear representations of the empty/mine/yours basis is strongly correlated with the strength of WSG. It has in 89% of the cases the same sign and a Spearman correlation of 0.850. Its correlation metrics are higher than those of all other baselines. We can further see, that the other features also correlate with WSG - although less. In 4a we can see how WSG success is linearly separable by the quality of board representations. In 4b we plot $X$ vs. $Y$ and see a strong monotonic relationship which is reflected in the high Spearman correlation. The cross-entropy loss before finetuning 4d is only monotonically related to WSG for the pairs where the strong model is already better before finetuning (i.e. right

side with $X > 0$). Neither the model size 4e nor the highly non-linear feature 4c has a clear relation with the success of WSG.

| X-Variable Definition | Sign Accuracy | Spearman $\rho$ | Max p-val (Spearman $\rho$) | $R^2$ |
|---|---|---|---|---|
| *Linear Probes* | | | | |
| Empty/Mine/Yours $\text{LP-Acc}(M_s) - \text{LP-Acc}(M_w)$ | **0.887 ± 0.057** | **0.850 ± 0.011** | **6.53e-29** | **0.298 ± 0.094** |
| Empty/Black/White $\text{LP-Acc}(M_s) - \text{LP-Acc}(M_w)$ | 0.821 ± 0.015 | 0.771 ± 0.023 | 1.13e-20 | 0.164 ± 0.061 |
| Empty/Filled $\text{LP-Acc}(M_s) - \text{LP-Acc}(M_w)$ | 0.806 ± 0.045 | 0.770 ± 0.014 | 1.91e-21 | 0.180 ± 0.067 |
| Linear × board % 3 $\text{LP-Acc}(M_s) - \text{LP-Acc}(M_w)$ | 0.458 ± 0.066 | -0.170 ± 0.121 | 7.52e-01 | 0.041 ± 0.059 |
| *Non interpretability based methods* | | | | |
| Cross-Entropy $\text{CE}(M_w, D_w) - \text{CE}(M_s, D_w)$ | 0.629 ± 0.036 | 0.674 ± 0.041 | 3.84e-14 | 0.188 ± 0.036 |
| N_parameters (n_p) $\log(\text{n\_p}(M_s)) - \log(\text{n\_p}(M_w))$ | 0.569 ± 0.085 | 0.067 ± 0.060 | 9.57e-01 | 0.023 ± 0.010 |

Table 2: **Metrics to predict PGR vs. actual PGR 5.1 5.1**. $N = 111$. Each row defines for each pair $(M_w, M_s)$ a value $X_i$, e.g. the first row is for the difference in accuracy for linear probes trained on $M_w$ and $M_s$. The correlation metrics on the right are computed between $X_i$ and $Y_i = PGR$ over all 111 pairs. The difference in accuracy of a linear probe, which predicts if a board field is empty or has the color of the current player or the other players color, is the most correlated with the success of WSG. The p-values for having at least as strong Spearman correlation as observed are for everything except the non-linear board state and number of parameters significant/very low - but we have to account for hierarchical dependence which inflates the number of independent datapoints (Bogdan, 2025). The full datapoints are visualized in 4. We display the mean and standard deviation over a sample of 3 independent runs. The p-values are the maximum across each run.

**Training dynamics.** In 5a we see that if the strong model surpasses its weak teacher, it happens early during the first 1000 finetuning steps. If the strong model only matched or did not reach the weak model's performance, the strong model reached its best validation score late in training. This suggests that if WSG occurs, it occurs through small changes to the model's parameters. But if small changes are not enough, it converges towards the weak model's output. Plot 5b supports this. In examples where WSG occurred the strong model at early-stopped-finetuning has roughly the same level of board representations as before finetuning. But in the examples of no WSG, its Othello board representations improved. These dynamics suggest that WSG succeeds when strong models already possess adequate world models and only need to adapt their output behavior, rather than learning fundamental representations during fine-tuning.

## 6 LIMITATIONS

We base our analysis on a toy language derived from the game of Othello, which may not transfer to frontier LLMs. Our 111 Othello-finetuning runs differ in model size and rule pairs, but they are based on seven different rules played on an Othello board. This hierarchical structure may inflate our correlation estimates, as model pairs from the same rule variants are not truly independent samples (Bogdan, 2025). Since we only use linear probes, we show that high-quality board representations correlate highly with WSG. However, we do not show that the strong model uses these features to fit the weak-finetuning signal. But, prior work (Nanda et al., 2023; Li et al., 2024a) has shown that board representations are used to play Othello. Our work provides insights into when WSG works, but it does not offer future-proof practical techniques, since probing for useful features in superhuman models might become very difficult.

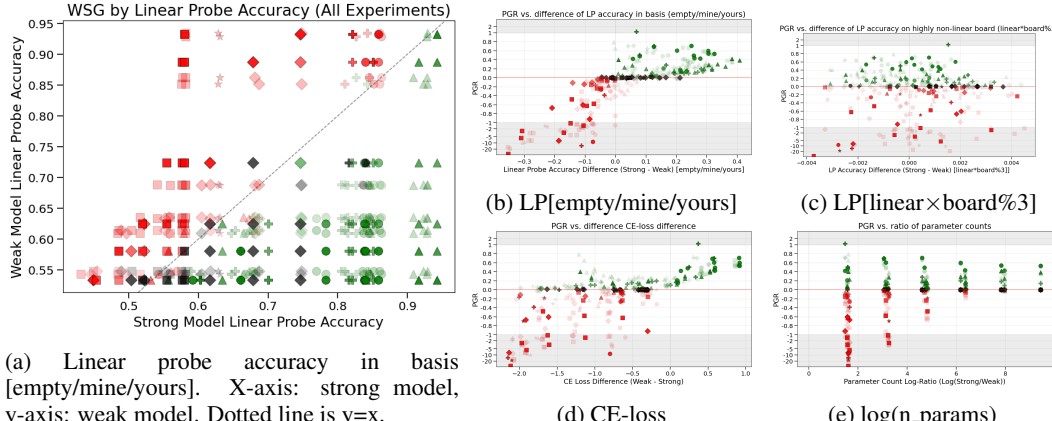

(a) Linear probe accuracy in basis [empty/mine/yours]. X-axis: strong model, y-axis: weak model. Dotted line is y=x.

(b) LP[empty/mine/yours]

(c) LP[linear×board%3]

(d) CE-loss

(e) log(n_params)

Figure 4: **Relation between PGR and different metrics.** Our goal is to find a metric $X_i(M_w, M_s)$ that given a pair of weak model $M_w$ and strong model $M_s$ can predict if WSG works. Finetuning $M_s$ through $M_w$ results in $M_{s \mapsto w}$. In all plots, the color represents the Performance-Gap-Recovered $Y_i = PGR$ 2.1 which measures how much WSG succeeded: Green means the strong model surpassed the weak model's performance on the weak task, i.e. $M_{s \mapsto w}$ plays legal moves in Othello more reliably than $M_w$. Black signifies that the strong model matched the performance, and red that it performed worse. The shapes represent the strong rule of the model. The x- and y-axis are different metrics based on $M_w$ and $M_s$. On the left 4a we plot strong model probe accuracy (x-axis) versus weak model probe accuracy (y-axis). The decision boundary that splits green vs. red points is naturally on the line LP-Acc($M_s$) > LP-Acc($M_w$). On the right, we plot $X_i$ on the x-axis vs. $Y_i = PGR$ on the y-axis (same data as in 2). In 4b, we can see how the difference in board representations in the basis [empty/mine/yours] shows a monotonic relationship with the PGR metric. The other metrics 4c, 4d, 4e are less related because there is no $X_i$ threshold on the x-axis that splits the green and red points and the relation between $X_i$ and $Y_i$ is weaker. We plot for a sweep of $n = 3$ the first sweep regularly and the second and third sweep at a high transparency.

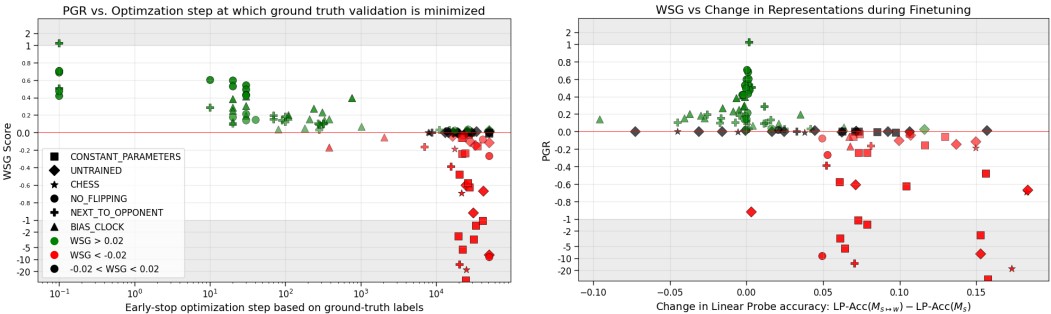

(a) PGR vs. optimization step at which the ground truth validation loss is minimized.

(b) PGR vs. the change in accuracy of board representations during finetuning. Positive means it got better.

Figure 5: **Finetuning dynamics of PGR and board representations.** The plot is similar to 4. We want to understand the board representations during finetuning. The left plot 5a has green points on the left and red points on the right. It indicates that if the strong model surpasses its weak teacher, this happens early in the finetuning. On the right 5b the green points are around $0$ and the red ones are positive. If WSG occurred, the board representations remain mostly unchanged. But if the strong model had worse representations and had to learn them during finetuning, WSG does not work.

# 7 CONCLUSION

While previous environments didn't use transformers, or leaked examples of the weak model task into the pretraining of the strong model, board games with different rules provide a clean environment. We show the first example of interpretable features that are related to the success of WSG.

## 8 REPRODUCIBILITY STATEMENT

Full information of our experiments can be found in the chapters 4.1 and 5.1 and the appendix A. Our code is reachable at `https://anonymous.4open.science/r/WSG_games-D22F` and we will make the data and all trained models accessible.

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

## A  TECHNICAL APPENDICES AND SUPPLEMENTARY MATERIAL

### A.1  MODEL HYPERPARAMETERS

Table 3: **Model hyperparameters.** We use GPT-2 style transformers (Radford et al., 2019) with $d_{\mathrm{mlp}} = 4 \times d_{\mathrm{model}}$ and $d_{\mathrm{model}} = n_{\mathrm{head}} \times d_{\mathrm{head}}$ through the TransformerLens library (Nanda and Bloom, 2022) and MinGPT (Karpathy, 2020) as used in (Li et al., 2024a). The `huge` transformer hyperparameters are identical to those of the model investigated in (Nanda et al., 2023; Li et al., 2024a).

| Model Size | Othello | | | | Tic-Tac-Toe | | | |
|---|---|---|---|---|---|---|---|---|
| | $n_{\mathrm{layer}}$ | $n_{\mathrm{head}}$ | $d_{\mathrm{model}}$ | $n_{\mathrm{parameters}}$ | $n_{\mathrm{layer}}$ | $n_{\mathrm{head}}$ | $d_{\mathrm{model}}$ | $n_{\mathrm{parameters}}$ |
| nano | 1 | 1 | 7 | $\approx 2.0\mathrm{K}$ | 1 | 1 | 1 | 68 |
| micro | 1 | 2 | 20 | $\approx 8.7\mathrm{K}$ | 1 | 2 | 4 | 390 |
| mini | 2 | 2 | 38 | $\approx 43\mathrm{K}$ | 2 | 4 | 8 | $\approx 2\mathrm{K}$ |
| small | 3 | 3 | 72 | $\approx 200\mathrm{K}$ | 3 | 4 | 16 | $\approx 10\mathrm{K}$ |
| medium | 4 | 5 | 140 | $\approx 970\mathrm{K}$ | 4 | 8 | 32 | $\approx 52\mathrm{K}$ |
| large | 6 | 6 | 264 | $\approx 5.1\mathrm{M}$ | 5 | 8 | 64 | $\approx 250\mathrm{K}$ |
| huge | 8 | 8 | 512 | $\approx 25\mathrm{M}$ | 6 | 16 | 512 | $\approx 1.2\mathrm{M}$ |

### A.2  TRAINING HYPERPARAMETERS

We do shorter pretraining (roughly 12h A100 vs. estimated 900h) than (Li et al., 2024a). As a worst-case comparison, on the largest standard Othello model, our approach results in an illegal move probability of 2.97% vs. 0.07% and a out of sample linear probe accuracy of 95.99% vs. 95.93%. However, all models we used in the paper appear fully converged since we only use smaller

models for standard Othello and the other rules are simpler with even smaller models playing close to perfect on them.

Table 4: **Othello: Hyperparameters for Pretraining, Finetuning, and Linear Probing.**

| Hyperparameter | Pretrain | Finetune | Linear Probe |
|---|---|---|---|
| Max epochs | 2 | 2 | 1 |
| Early stop patience | — | 100 | 2 |
| Early stop val every n steps | — | 100 | 100 |
| Batch size | 512 | 512 | 32 |
| Weight decay | 0.1 | 0.1 | 0.01 |
| Learning rate | $5 \times 10^{-4}$ | $1 \times 10^{-5}$ | $1 \times 10^{-4}$ |
| Adam betas $(\beta_1, \beta_2)$ | (0.9, 0.95) | (0.9, 0.95) | (0.9, 0.95) |
| Grad norm clip | 1.0 | 1.0 | — |
| LR decay schedule | Cosine | — | — |
| LR warmup | First 5% (linear) | First 5% (linear) | First 5% (linear) |

Table 5: **Tic-Tac-Toe: Hyperparameters for Pretraining, Finetuning.**

| Hyperparameter | Pretrain | Finetune |
|---|---|---|
| Learning Rate | $1 \times 10^{-3}$ | $1 \times 10^{-5}$ |
| Weight Decay | $1 \times 10^{-4}$ | $1 \times 10^{-2}$ |
| Max Epochs | 1000 | 1000 |
| Batch Size | 64 | 64 |
| Early Stopping Patience over epochs | 3 | — |
| Early Stopping Patience over optimization steps | — | 100 |

