# OpenReview forum: "Predicting Weak-to-Strong Generalization from Latent Representations"
_ICLR.cc/2026/Conference — Submitted to ICLR 2026_

### Official Review · Reviewer_iwNH · 2025-10-17

**Soundness:** 2
**Presentation:** 3
**Contribution:** 2
**Rating:** 4
**Confidence:** 4

**Summary:**

This paper introduces a clean, contamination-free toy testbed using variations of Othello and Tic-Tac-Toe rules for Transformer models. The authors empirically support the hypothesis that WSG success is directly tied to the strong model's ability to leverage its superior internal features, specifically its board representations. They quantify this mechanism by demonstrating a strong Spearman $\rho$ of $0.85$ between WSG success and the superiority of the strong model's representations, as measured by linear probes.

**Strengths:**

- The experiments are interesting. The use of linear probing (or more broadly, mechanistic interpretability methods) in Weak-to-Strong Generalization (WSG) is novel. To my best knowledge, this is the first paper to investigate this direction.
- The motivation is clear. The claim regarding the lack of explanatory power concerning representations in related work is, to my knowledge, accurate. Most previous WSG theories utilize linear models and the Gaussian feature assumption to derive conclusions, including the statement that "WSG works when the strong model learns how to leverage its superior features." While this claim has appeared in many related WSG works, this paper's effort to address the absence of a mechanistic explanation is appreciated.
- The paper is generally well-written with intuitive figures and explanations. For instance, highlighting "train-test contamination" as one of the paper's advantages is excellent. This specific limitation appeared in the original WSG paper but seems to have been neglected by much of the subsequent work in this field.

**Weaknesses:**

**Firstly, let me say that if my concerns are addressed completely and satisfactorily, I would be happy to revise my review and increase my rating.** It is possible that I have not fully grasped the setting of the board game variations; if so, please feel free to correct my understanding. Aside from the paper's experiments, I genuinely appreciate the underlying thinking presented. If the paper could delve deeper into the questions it raises, I would strongly advocate for a high score. However, for now, I feel there are areas where the paper could be improved and explored more thoroughly. Therefore, I need to reach a mutual understanding with the authors on several points, which requires a more detailed discussion.

## 1. Problem Setting
This setting appears to differ from the standard WSG setup. Figure 1 suggests that the weak model is trained with a basic representation, while the strong model is trained with a better representation. However, in a practical WSG setting, the weak model is typically trained on ground truth data before its predictions are used to fine-tune the strong model. This would imply that the weak model's training source is "better" than the strong model's fine-tuning data (which is derived from the weak model). I am slightly confused by the assignment of representation quality for the weak and strong models in your proposed testbed compared to the typical WSG data pipeline.

## 2. How Does Linear Probing Support Your Claim?
**I am not convinced that linear probing adequately explains the mechanism by which the strong model leverages superior features to achieve WSG.** I would argue that if the WSG phenomenon occurs, and if you are using linear probing to measure the quality of representations, it is almost inevitable that the model is utilizing better features. Therefore, your experiments appear logically constrained to showing that, under the two specific settings you consider, WSG can occur under certain parameter configurations. Intuitively, if WSG occurs, your experimental results are guaranteed to follow the observed trend. Conversely, if WSG does not occur, the trend will not appear. As a result, the strong correlation you found is expected and not surprising. To my knowledge, linear probing can only indicate whether a representation encodes useful information about the task, but it cannot explain why this is the case. Consequently, the paper does not clearly explain why the strong model acquires these beneficial features in the first place. **If your main contribution is dealing with the verification of the previous statement in WSG that "WSG works when the strong model learns how to leverage its superior features," then I believe that the linear probing approach is insufficient to verify "how."** This is because linear probing has been widely employed in different tasks, and it is well-established that the performance of the final task can be predicted by probing the representations. In this sense, this paper merely validates a widely-acknowledged conclusion in a specific setting—WSG—which seems trivial and superfluous. However, if the paper's main goal is to apply interpretability tools to WSG, then it is easier for a researcher in WSG to accept the contribution.

What exactly constitutes a "better representation" or "superior features" in the context of your work? I strongly encourage the authors to delve deeper into this fundamental question. For instance, the related WSG paper by Ildiz et al. (2025) you cited characterizes the condition under which WSG occurs using eigenvalue analysis. Also, Dong et al. (2025) uses intrinsic dimension. Therefore, I suggest the authors broaden the scope of this paper beyond just accuracy and linear probing to offer a more profound definition and evaluation of representation quality (for example, eigenvalue and intrinsic dimension).

Furthermore, I believe that generally, if the final task accuracy is high, the features are likely good, but the converse—if the features are good, the final task accuracy is not necessarily high—is a more subtle point. The paper's finding of a trend between representation superiority (linear probe accuracy) and WSG success (final task accuracy) effectively suggests that representation quality can predict the occurrence of WSG. This shifts the core question to: **what is the precise relationship between linear probing accuracy and the final task accuracy?** Verifying this relationship seems challenging because the linear probing literature involves subtle decisions regarding which layer's representation to use and how to extract representations (especially in NLP), potentially requiring extensive comparative experiments. Your conclusion would be significantly more convincing if you could demonstrate two points: (1) The correlation between linear probing accuracy and final task accuracy is typically much weaker in other tasks; and (2) This correlation is significantly strong under the specific WSG settings you consider.

## 3. Minor Points

Typos/Formatting:
- Line 217: Reference should be to Appendix A.1.
- Line 266: Reference should be to Figure 2.
- Line 335: The variables round_down, n_layer, and d_model should be formally explained. While they are easy to understand, variables derived from code that appear in the main text must be defined.
- Line 400: Change replicated "5.1" to something more formal. Also, use the LaTeX formula $N=111$ instead of N=111 for consistency.
- Line 402: The phrase "2.1" here is unclear and too casual.

Related work:
- Missing references [1-4] concerning the theoretical analysis of weak-to-strong generalization.
- Although the paper mentions the concept of "The Linear Representation Hypothesis," (Line 133) it does not even cite the "Linear Representation Hypothesis" paper [5]. I understand that the core idea behind the linear representation hypothesis has been investigated for some years before [5], which may explain why you did not cite it. However, to my knowledge, the specific term "Linear Representation Hypothesis" first appeared in [5].
- While linear probing has been extensively used in LLMs in recent years, Section 2.2 of the paper does not include any citations from 2024. I encourage the authors to update this subsection by citing some recent, relevant papers published beyond 2023 on linear probing or the Linear Representation Hypothesis in LLMs, such as [6-10], to better support the context of this work and demonstrate awareness of current literature.

[1] Relating Misfit to Gain in Weak-to-Strong Generalization Beyond the Squared Loss. ICML 2025.

[2] The Capabilities and Limitations of Weak-to-Strong Generalization: Generalization and Calibration. arXiv:2502.01458.

[3] From Linear to Nonlinear: Provable Weak-to-Strong Generalization through Feature Learning. HiLD 2025.

[4] On the Mechanisms of Weak-to-Strong Generalization: A Theoretical Perspective. arXiv:2505.18346.

[5] The Linear Representation Hypothesis and the Geometry of Large Language Models. ICML 2024.

[6] Representation Engineering: A Top-Down Approach to AI Transparency. arXiv:2310.01405

[7] Inference-Time Intervention: Eliciting Truthful Answers from a Language Model. NeurIPS 2023.

[8] On the Origins of Linear Representations in Large Language Models. ICML 2024.

[9] Towards Tracing Trustworthiness Dynamics: Revisiting Pre-training Period of Large Language Models. ACL Findings 2024.

[10] When is Task Vector Provably Effective for Model Editing? A Generalization Analysis of Nonlinear Transformers. ICLR 2025.

**Questions:**

1. Given a fixed parameter for the strong model in Figure 2, a stronger weak model results in a lower PGR. The authors explain this phenomenon by stating that "For large weak models (medium, huge), PGR is negative, because they are already playing close to perfect." I find the observation that PGR is negative counter-intuitive, even if the weak model is highly reliable. While I agree the improvement over the weak model should be smaller, it should not lead to further degradation (a negative gain). This observation is peculiar and requires further discussion.
2. Why is there a very large variance in Figure 2?
3. Why is $R^2 = 0.298$ such a small number that might not indicate a significant trend? I understand that other metrics may sufficiently indicate the trend, but why is the $R^2$ value so low?
4. Line 71: The claim "We analyse the toy task of Othello, because it is the first truly pretrain leakage free transformer environment for WSG" requires justification. Why is it the first? The authors make this strong statement without proper explanation or citation.
5. Line 188: The claim "Our mechanistic interpretability approach provides the first direct empirical validation of these theoretical predictions using realistic transformer architectures." I want to challenge the word **"first"** in this sentence (and in some other positions in this paper) and remind you that the paper by Xue et al. (2025) you cited also developed a representation-based metric that strongly correlates with W2SG performance. Given that linear probing is also a representation-based metric, would you like to provide a more detailed discussion (or potential experimental comparison) between the two?

---

> ### Author Response · Authors · 2025-12-03
>
> Review 3:
> Thank you for your reviews and detailed corrections.
>
> 1. Problem setting
> Most WSG experiments use two models pretrained on the same dataset and then train the weak model on the ground-truth data of a specific task. The strong model gets finetuned and its performance is measured on this new task. Works such as [1] hypothesise that the strong model is able to surpass its weak teacher on the new task, because during the pretraining the strong model was able to achieve better representations (due to its size, or in some setups different data). We are constructing an example that fulfils the premises of this setup by training a weak model on regular Othello and a strong model on a rule variation that also requires board representations of the same environment as the weak task.
>
> 2. How Does Linear Probing Support Your Claim?
> In e.g. Figure 2 the dotted lines represent the performance of the strong model on the weak task before finetuning. It is in almost all cases worse than that of the weak model (PGR <= 0). Yet the board representations (as measured by linear probes) in Figure 4 are in part better than that of the weak model (there are points right/under the x=y line). This shows that the linear probe is not predicting the performance on the weak task. An interpretation is that since the strong model was trained on a different rule it has great representations, but still fails at the weak task, since it utilizes the representations differently. However, after finetuning through the weak model, the strong model outperforms the weak model. Our statement is that this behaviour is strongly correlated with the strength of board representations (before finetuning) and independent from the performance of the strong model on the weak task before finetuning. An interpretation is that the strong model learned from the weak model how to utilize its superior understanding of the board to generalize from the provided weak examples how the weak rule works - and then outperforms its weak supervisor.
>
> Table 2 displays an ablation in the same spirit as your proposed experiment. It shows that probing for board features that are not aligned with the weak task is not/less correlated with the final performance. E.g. probing for the non-linear 'Linear x board % 3' transformation of the Othello board has much lower predictive power for the final performance compared with probing for the actual board state (0.850 vs. -0.170 Pearson correlation).
>
> 3. Minor points & related work
> We updated these in our paper. Much appreciated!
>
> Questions:
> 1. A negative PGR means that the strong model did not surpass its weak teacher during finetuning. The strong model still improves during finetuning, just never surpasses the performance of the weak teacher.
> 2. One of the 10 training runs failed to converge, but to keep results unbiased we did not rerun it.
> 3. R2 is a metric for the linear relationship between two metrics. Spearman correlation measures a monotonic relation. Since Spearman correlation is high, but R2 (i.e. Pearson Correlation) is low, we can deduce that there is a monotonic (but not linear) relationship. This is visible in Figure 4 (b), where the dots go from the bottom left to the top right (monotonic), but in a very non-linear fashion since the y-axis was log-transformed.
> 4. All prior transformer setups of WSG use two transformers pretrained on the same dataset. Our unique idea of using two different game rules instead of the same pretrain dataset avoids this leakage.
> 5. Thank you, we added it to our paper. Note that the metric from [2] is PCA based and not interpretable and the experiment setup suffers from pretrain leakage. It is therefore unclear if the metric would work if the strong model has not seen high-quality examples of the weak task. Additionally our study provides more experiments with consistent evidence. Still, we decided to soften our statement by removing the "the first" from our abstract.
>
> [1] Collin Burns et al. Weak-to-Strong Generalization: Eliciting Strong Capabilities With Weak Supervision.
> [2] Yihao Xue et al. Representations Shape Weak-to-Strong Generalization: Theoretical Insights and Empirical Predictions

---

### Official Review · Reviewer_niCS · 2025-10-30

**Soundness:** 3
**Presentation:** 4
**Contribution:** 2
**Rating:** 4
**Confidence:** 4

**Summary:**

The paper demonstrates weak to strong generalization for transformers in a clean game environment (Othello and Tic-Tac-Toe). Furthermore, empirically this work confirms the already known hypothesis that WSG occurs if and only if the strong model has better representations (or superior features) of the target task than the weak model itself.


Decision: My actual score for this paper would be 5. I’m okay with either decision for this paper.

**Strengths:**

1) The paper provides a better empirical evidence of the following known hypothesis in a clean game environment: WSG occurs iff strong model has a better representation of the target task than the weak model. These empirical findings therefore give further confidence into our current primarily theoretical understanding of WSG. The paper provides empirical evidence of this hypothesis for transformers -- which was not done in the prior work.
2) The paper is clean and very well written.

**Weaknesses:**

1) My primary weakness of this paper is the motivation itself. There are various theoretical results already showcasing good understanding of WSG, these results work more generally including transformers. Although the results in the paper are interesting, I feel there is no new understanding that can be generated from this work which was not already known.

2) Also for a completely new complicated task, it is unclear on how to decide whether the strong model has learnt the better representations of that task.

**Questions:**

Let’s say I have a new task — how can we determine whether the model has actually learned better representations? Is there a principled way to assess whether the model has learned a good representation for a given task?

---

> ### Author Response · Authors · 2025-12-03
>
> Review 2:
> Thank you for your thoughtful review.
>
> Motivation:
> Yes, our paper is providing the first empirical evidence for a hypothesis that has been theoretically analysed before under mathematical assumptions. WSG has been investigated in many papers in the last 2 years, without fixing the pretrain leakage. Previous papers often state the (theoretically justified) hypothesis even though there is not enough evidence for it. We solved both problems in our paper (by providing empirical evidence in a clean toy-environment and proposing a way of modifying popular existing game based setups).
>
> Representations of new task:
> To my knowledge there is unfortunately no such generally applicable way. We picked game environments because it is simpler to measure the strength of representations. Our paper tries to improve our fundamental understanding of WSG and we do not claim a working technique that can be applied broadly. One speculative (!) interpretation could be that if AI is more capable, but sufficiently close to humans, it is capable of understanding weak examples of aligned data without overfitting on these. E.g. if we provide examples of safe but simple code it can learn to write safe but superhuman code.

---

### Official Review · Reviewer_N2kT · 2025-10-31

**Soundness:** 2
**Presentation:** 3
**Contribution:** 2
**Rating:** 2
**Confidence:** 3

**Summary:**

This paper studies weak-to-strong generalization (WSG) in which a weak model produces labels used to train a strong model. It has been empirically shown in frontier LLMs that the strong student can surpass its weak teacher. Here the authors propose a new toy testbed, where they train transformers on different rule variants of two games, Othello and Tic-Tac-Toe, to define weak & strong models. They claim to provide the first empirical evidence that WSG success depends on the stronger model’s internal representation quality, reporting a high Spearman correlation of 0.85 between WSG success and linear-probe accuracy on board-state representations. The paper concludes a proof-of-concept that strong models outperform weak teachers if and only if they possess such superior features.

**Strengths:**

* The problem has clear motivation, and most parts of the paper are well-written, making the presentation easy to follow.
* Code is anonymously open-sourced with a clean and reproducible setup that shows strong correlational results.
* Though investigated through a toy framework, the problem is important as AI models are becoming increasingly stronger, and understanding their internal representation potentially helps facilitate alignment.

**Weaknesses:**

* **Overstated Novelty**: Most of the empirical evidence from this toy testbed is correlational in this paper, but at times the wording (e.g. the "if and only if" statement in the abstract) seems to imply that the superior features learned by the strong enable it to achieve WSG. The only insight from the study is that better WSG and better representations tend to co-exist, but the bi-directional sufficiency/necessity and causal relationships are **NOT** adequately justified (via ablations or other methods). With this being said, this proof-of-concept for "when WSG succeeds" appears weak, as real WSG involves much richer internal/latent dynamics than toy board states can model. While the authors did point out this limitation, I believe it ultimately makes this paper lack depth and rigor.

* **Inconsistency with Existing Work**: The main conclusion only revolves around the **strong** representations of the strong model, but previous work [2] has demonstrated that in large models, WSG involves an intricate interaction between the representations of both the weak and strong models, and WSG will not be as effective if this interaction causes the strong model to replicate the weak model's inability. This nuance is entirely missing here possible due to oversimplified toy setting and shows an important gap. In practice, strong representations of certain features could cause overfitting and amplify the weak model's bias. This does not manifest in the considered testbed.

* **Inaccuracy in Problem Setup**: While the framework proposed in the paper can prevent pretrain leakage and train-test contamination to some extent, I believe the following two issues persist. (1) The claim in Section 2.1 is inaccurate: most WSG studies focus on two models pretrained similarly (covering similar scope of tasks). For example, the original OpenAI paper [1] used GPT-2 and GPT-4, and the superiority of the strong model mainly derives from richer representations across general tasks. Here the two transformers are trained on different rule variants and less representative of practice. (2) The setup seems to reduce obvious contamination, but it may not fully rule out pretraining leakage at the representational level. Both models still share similar "token spaces", and more justifications might be needed here.

[1] Collin Burns et al. Weak-to-Strong Generalization: Eliciting Strong Capabilities With Weak Supervision.

[2] Yihao Xue, Jiping Li, and Baharan Mirzasoleiman. Representations Shape Weak-to-Strong Generalization: Theoretical Insights and Empirical Predictions. ICML 2025.

**Questions:**

Most questions are raised in the Weakness section. Regarding the setup, I have two short follow-up questions:

1. Have you tested whether manually degrading the board representations reduces WSG success, which can potentially support some causality?

2. As already pointed out in the paper, the setup may not directly generalize to frontier models. Yet, what could be some extensions from the toy framework?

---

> ### Author Response · Authors · 2025-12-03
>
> Review 1:
> Thank you for your detailed review
> Overstated Novelty:
> We softened the formulations to "In Othello, the strong student model surpassing the weaker teacher is strongly correlated with having better board representations." and "we provide empirical evidence for this on transformers." (removing the "if and only if" and "the first")
>
> Inconsistency with Existing Work:
> The phenomenon of a strong model overfitting on weak model appears in our experiment setup in all runs when a strong model fails to improve over its weaker teacher but reaches PGR=0, i.e. exactly the same performance as the weak model (Fig 2).
> WSG occurs if the strong model surpasses the performance of its weak teacher - before eventually overfitting to the weaker supervision signal. Our statement is that this occurs if and only if the strong model has better board representations. This is consistent with prior work.
>
> Inaccuracy in Problem Setup:
> In our analogy the game of Othello (and such the token space) represent the real-world environment. Both AI and humans share the same underlying reality as well. Our setup is the first truly pretrain leakage transformer based environment and to achieve zero pretrain leakage we decided to choose Othello and its different rules. Our experiment setup contains all of the key properties that were previously deemed important (2 models with different tasks, the weak model finetunes the strong one), but we were able to remove the pretrain leakage that existed so far by using games in a different way than e.g. [1]
>
> Did you degrade board states?
> No, we did not degrade board representations. Instead we created models with different levels of board representations by crafting Othello-rules that depend to different degrees on the board state.
>
> Extensions from toy setup:
> The novelty of our setup is the idea of using games with different rules to avoid pretrain leakage. This could be taken further by e.g. using video games instead of board games.
>
> [1] Collin Burns et al. Weak-to-Strong Generalization: Eliciting Strong Capabilities With Weak Supervision.

---

### Meta-Review · Area_Chair_qR18 · 2026-01-10

**Summary:**

This paper studies the weak-to-strong generalization using some toy settings. The empirical results are extensive but somewhat expected. I find it less informative in the sense that it didn't provide additional insights for us to understsand weak-to-strong generalization.

After reading the paper and review, I find this paper falls short at (1) limited insights from the empirical results; (2) overclaimed novelty; and (3) over-simplified experimental settings. Combining reviewers' opinions, I hence recommand rejection.

**Reviewer Concerns:**

Most concerns are about the limited motivation/novelty and the experimental settings. The authors didn't fully address these concerns.

**Reviewer Scores:**

The final score is 4,4,2, and no one changed the score.

---

### Decision · Program_Chairs · 2026-01-26

Reject